# *PRKAR1A* and Thyroid Tumors

**DOI:** 10.3390/cancers13153834

**Published:** 2021-07-30

**Authors:** Georgia Pitsava, Constantine A. Stratakis, Fabio R. Faucz

**Affiliations:** 1Division of Intramural Population Health Research, Eunice Kennedy Shriver National Institutes of Child Health and Human Development, National Institutes of Health, Bethesda, MD 20892, USA; georgia.pitsava@nih.gov; 2Section on Endocrinology and Genetics, Eunice Kennedy Shriver National Institute of Child Health and Human Development, National Institutes of Health, Bethesda, MD 20892, USA; stratakc@mail.nih.gov

**Keywords:** thyroid carcinoma, *PRKAR1A*, PKA, Carney complex, cAMP

## Abstract

**Simple Summary:**

In 2021 it is estimated that there will be 44,280 new cases of thyroid cancer in the United States and the incidence rate is higher in women than in men by almost 3 times. Well-differentiated thyroid cancer is the most common subtype of thyroid cancer and includes follicular (FTC) and papillary (PTC) carcinomas. Over the last decade, researchers have been able to better understand the molecular mechanisms involved in thyroid carcinogenesis, identifying genes including but not limited to *RAS*, *BRAF*, *PAX8/PPARγ* chromosomal rearrangements and others, as well as several tumor genes involved in major signaling pathways regulating cell cycle, differentiation, growth, or proliferation. Patients with Carney complex (CNC) have increased incidence of thyroid tumors, including cancer, yet little is known about this association. CNC is a familial multiple neoplasia and lentiginosis syndrome cause by inactivating mutations in the *PRKAR1A* gene which encodes the regulatory subunit type 1α of protein kinase A. This work summarizes what we know today about *PRKAR1A* defects in humans and mice and their role in thyroid tumor development, as the first such review on this issue.

**Abstract:**

Thyroid cancer is the most common type of endocrine malignancy and the incidence is rapidly increasing. Follicular (FTC) and papillary thyroid (PTC) carcinomas comprise the well-differentiated subtype and they are the two most common thyroid carcinomas. Multiple molecular genetic and epigenetic alterations have been identified in various types of thyroid tumors over the years. Point mutations in *BRAF*, *RAS* as well as *RET/PTC* and *PAX8/PPARγ* chromosomal rearrangements are common. Thyroid cancer, including both FTC and PTC, has been observed in patients with Carney Complex (CNC), a syndrome that is inherited in an autosomal dominant manner and predisposes to various tumors. CNC is caused by inactivating mutations in the tumor-suppressor gene encoding the cyclic AMP (cAMP)-dependent protein kinase A (PKA) type 1α regulatory subunit (*PRKAR1A*) mapped in chromosome 17 (17q22–24). Growth of the thyroid is driven by the TSH/cAMP/PKA signaling pathway and it has been shown in mouse models that PKA activation through genetic ablation of the regulatory subunit *Prkar1a* can cause FTC. In this review, we provide an overview of the molecular mechanisms contributing to thyroid tumorigenesis associated with inactivation of the *RRKAR1A* gene.

## 1. Introduction

### 1.1. Incidence of Thyroid Cancer

Thyroid cancer is the most common endocrine tumor in the general population and the incidence continues to rise in the United States [1]. The American Cancer Society estimates that there will be 44,280 new cases of thyroid cancer (12,150 in men and 32,130 in women) and about 2200 deaths (1050 in men and 1150 in women) in the United States in 2021 [2]. The increased incidence could be possibly attributed to the increased detection of these tumors with imaging technics (like ultrasound and computed tomography (CT)) that better characterize incidental findings of small thyroid nodules [3].

### 1.2. Subtypes of Thyroid Cancer

In the majority of patients (about 90%), well-differentiated epithelial thyroid cancer is present; this is further categorized into papillary thyroid cancer (PTC) and follicular thyroid cancer (FTC), based on histological criteria [3,4]. The long-term survival of those patients is excellent, with 5-year relative survival rate (as of 2010–2016) being as high as 98% in all stages (>99% for local tumors and 55% for tumors with distant metastases) [2]. However, FTC tends to behave more aggressively with distant metastases and vascular invasion [5,6] being more common and thus its prognosis is poorer than PTC [7]. The rest of the thyroid carcinomas (~2–3%) include medullary thyroid carcinomas (MTCs) that originate from the calcitonin-producing parafollicular C cells, while anaplastic carcinomas (ATCs) and poorly differentiated carcinomas account for the remaining 7–8% [4]. In addition to the above tumors, benign thyroid tumors that usually present as thyroid nodules as well, include benign hyperplasia or benign follicular adenomas [3].

### 1.3. Evaluation of a Thyroid Nodule

Thyroid nodules are quite common and are found either clinically or as an incidental finding on imaging studies [8]. The majority of them are benign [9]; only a small percentage harbors thyroid cancer [8]. The initial steps in the evaluation of a thyroid nodule consist of medical history including symptoms (recent onset of hoarseness, neck discomfort or dysphagia), history of head/neck radiation and personal/family history of cancer, followed by physical examination and measurement of serum thyrotropin levels. Ultrasonography (US) is the next step in order to determine the size of the nodule, its characteristics and to assess for cervical lymphadenopathy [10]. If thyrotropin levels are normal or elevated and the nodule size is >1 cm, then fine needle aspiration (FNA) is indicated, according to the American Thyroid Association guidelines [11,12]. If thyrotropin levels are low, then Iodine-123 or technetium-99m thyroid scanning is recommended. In the case that the nodule is nonfunctioning and bigger than 1cm, FNA is the next step. If the cytological interpretation is benign, then repeated FNA is not required unless suspicious features appear in the follow up [12,13]. Currently, US-guided FNA is the gold standard in the diagnosis; however, in about 25% of the cases, the diagnosis remains indeterminate [9,14,15,16,17,18,19,20]. If cytologic results are interpreted as atypia of underdetermined significance or follicular lesion of underdetermined significance, then in the case of high suspicion, assessment of the aspirate for molecular abnormalities (e.g., mutations or rearrangements) is indicated [21].

### 1.4. Thyroid Cancer as Part of Genetic Syndromes

Thyroid malignancies are also associated with at least two syndromes with inherited tumor predisposition, Cowden syndrome (CS, OMIM# 158350) and Carney Complex (CNC, OMIM #160980). CS is a multiple hamartoma syndrome, including FTC, brain and breast cancer. It is caused by inactivating mutations in the *PTEN* gene, a dual-specificity phosphatase that negatively regulates PI3 Kinase/AKT pathway; mutations in this gene have been detected in 5% of FTCs [22]; however, a mouse harboring a deletion of *Pten* in the thyroid developed thyroid hyperplasia and not FTC [23].

In this review, we will focus on CNC, which is a multiple neoplasia syndrome that presents as the complex of myxomas, spotty skin pigmentation and endocrine tumors (Table 1) [24]. CNC is caused by inactivating mutations in the *PRKAR1A* gene (mapped in 17q22–24) [25]; somatic mutations in this gene have been reported in sporadic cases of thyroid cancer [26]. In tumors associated with CNC as well as in thyroid and adrenal tumors with downregulation of *PRKAR1A*, allelic losses of the 17q22–24 *PRKAR1A* chromosomal locus are frequently identified and are associated with changes in PKA activity [26,27,28,29].

## 2. *PRKAR1A* Structure and Function

Cyclic adenosine monophosphate (cAMP)-dependent protein kinase type 1-alpha regulatory subunit is encoded by the *PRKAR1A* gene. *PRKAR1A* consists of 11 exons; ten of them (2–11) are coding. Protein kinase A (PKA) (Figure 1), a serine/threonine kinase, is a second messenger-dependent enzyme and it is involved in G-protein coupled intracellular pathways. It is the main mediator of cAMP actions for various cellular processes in mammals, including cell differentiation, proliferation, and apoptosis [30,31,32].

The PKA holoenzyme is a hetero-tetramer composed of two regulatory (R) subunits and each is bound to one catalytic (C) subunit [33]. Four subtypes of R (RIα, RIβ, RIIα, RIIβ) and four subtypes of C (Cα, Cβ, Cγ and Prkx) subunits have been identified so far. A gene is coding each R (*PRKR1A*, *PRKR1B*, *PRKR2A*, *PRKR2B*) and each C (*PRKACA*, *PRKACB*, *PRKACG*, *PRKX*) subunit, respectively [33,34]. Two major isozymes have been identified, type I and type II PKA, based on their chromatographic elution patterns [32]; they are comprised of homodimers of either RIα and RIβ or RIIα and RIIβ, respectively [31,35]. In the basal state, the catalytic subunits bind mostly to type II subunits [31,35,36,37]. When cAMP binds to the R subunits, it alters their conformation; this causes the dissociation of each active C subunit from the dimer with the corresponding R subunit. Following that, the free C subunits phosphorylate threonine and serine residues of proteins that are critical to the activation of downstream processes [38,39,40].

RIα haploinsufficiency, as shown by mice and human studies, predisposes to the development of tumors [29,41]. The majority of *PRKAR1A* mutations result in premature stop codons with unstable mRNAs undergoing nonsense-mediated decay [25,42]. In the thyroid, PKA through the production of cAMP, signals downstream of thyrotropin (TSH) on cell proliferation and differentiation; increased levels of TSH in humans have been associated with thyroid tumors [43]. In addition, in a series of thyroid tumors, the Cα subunit was investigated but no mutations were detected [44].

## 3. Role of *PRKAR1A* in Thyroid Cancer

### 3.1. Mouse Studies

#### 3.1.1. Mouse Models

In 2004, Griffin et al. generated a mouse model carrying an antisense transgene for *Prkar1a* exon 2 (X2AS) under the control of a tetracycline responsive promoter (the Tg(Prakr1a*x2as)1Stra, Tg(tTAhCMV)3Uh, or tTA/X2AS) [45]. Increased cAMP signaling was demonstrated due to significant *Prkar1a* downregulation and the mice exhibited a more severe phenotype with high incidence of thyroid lesions (thyroid follicular hyperplasia and adenomas). This was an extremely rare finding in wild type animals but quite common in those with the genetic defect (as it is common among patients with CNC). Furthermore, the lesions were associated with allelic loss of the *Prkar1a* locus on chromosome 11 as it happens in thyroid tumors with *PRKAR1A* mutations. Moreover, tumor tissues demonstrated an increase in the activity of type II PKA and higher RIIβ levels, an abnormal cAMP response.

In a later study, *Prkar1a* haploinsufficiency in mice was investigated. It was shown that *Prkar1a* haploinsufficiency leads to tumor development arising in cAMP-responsive tissues, including among others, benign and malignant thyroid neoplasms [41]. Mice heterozygous for a conventional null allele of *Prkar1a* (*Prkar1a*^Δ2/+^ mice) were generated. These mice developed tumors in the same spectrum as CNC patients. Thyroid neoplasms were present in 10% of *Prkar1a*^Δ2/+^ mice [41]. In addition, allelic loss occurred in a portion of tumor cells, as indicated by genetic analysis, suggesting that complete loss of *Prkar1a* plays a vital role in tumor formation.

A different mouse model, carrying a thyroid-specific deletion of *Prkar1a* (Tpo-R1αKO) was studied [46]. In 43% of mice, FTC was observed by 1 year of age. However, distant hematogenous metastases were not present, which is a key feature of FTC in humans [46]; this could potentially suggest that metastases may be triggered by another genetic mutation in the case of *Prkar1a* mutation in the thyroid. An interesting observation by the authors was that thyroid ablation of *PRKAR1A/Prkar1a* is the only genetic change that has been described that results in FTC in both mice and humans.

#### 3.1.2. Activation of mTOR Pathway

The role of PKA as a key regulator of FTC has also been suggested by a recent study demonstrating a concurrent activation of PKA and mTOR. In this study a double *Prkar1a*-*Pten* knockout mouse (*DRP*-*Tpo*KO mice) with thyroid-specific deletion of both genes was generated and was compared to signaling alterations to human FTCs [1]; they found that mice developed aggressive FTC that exhibited 100% penetrance by 8 weeks of age. In addition, well-differentiated lung metastases appeared to be common in these mice (approximately one third of them), mimicking the human disease. The signaling pathways were analyzed and it was shown that PKA and the mammalian target of rapamycin (mTOR) pathways were consistently activated. mTOR has an essential role in promoting the metabolic changes that occur during tumorigenesis and is regulated by the AMP-dependent protein kinase (AMPK) [47]. AMPK is activated under nutrition restriction or increase in the AMP/ATP ration in order to increase energy production [48].

It has been suggested before that mTOR could be activated by *Prkar1a* deletion and that it could possibly interact with Prkar1a directly [49], but the data remain controversial [50]. Furthermore, activation of mTOR by TSH has been suggested to be partly due to PKA phosphorylation of the target of rapamycin complex 1 complex member PRAS40 [51]. Mouse models have been developed over the years that recapitulate how human FTCs progress from benign follicular adenoma (at one year of age) in the *Pten-TpoKO* [23] to locally invasive FTC as in the *R1a-TpoKO* [46] and subsequently to invasive and distantly metastatic FTC. The authors identified PKA and mTOR as essential signaling pathways and showed that activation of mTOR can occur independently of Akt [1]. Further, the concurrent activation of PKA and mTOR that was observed in human FTCs led to the conclusion that PKA activates mTOR/p70S6K that results in thyroid cancer, indicating that PKA is a vital component regulating FTC in both mice and humans [1].

The same group reported that, in FTCs, both in mice and humans, AMPK and mTOR pathways are activated concomitantly [52]. They showed that the tumor suppressor that causes Peutz–Jeghers syndrome, LKB1, mediates the signaling from PKA to AMPK in driving tumorigenesis [53,54]. The role of AMPK in the development of cancer has not been determined yet; according to the literature, it can act either as tumor promoter or tumor suppressor [55,56]. LKB1, like AMPK, can act as tumor promoter/suppressor as well, depending on the context [55,57,58,59,60,61]. Even though it typically suppresses the activity of mTOR [56,62], there is evidence that it can also act as a tumor promoter [63,64,65], which means that its functions depend on the type of tissue and other intracellular signals that may be present.

#### 3.1.3. Targeting Downstream Effectors of cAMP

Because of the various effects of cAMP in physiological responses, therapies targeting cAMP signaling result in side effects; thus, understanding downstream targets of cAMP signaling has been attempted in a number of studies [66,67]. The roles of Rap1 and Epac1 in *Prkar1a*-associated thyroid carcinogenesis have been studied [68]. *Rap1* is a small GTPase essential for effective signal transduction. There are two isoforms and each one is encoded by a separate gene, *Rap1a* and *Rap1b*, respectively. The activity of Rap1 has been shown to be regulated by both PKA and cAMP though signaling by TSH [69]. Increased Rap activity has been linked to various cancers, including thyroid cancer, while dysregulation of Rap1 has been postulated to contribute to the development of malignancy [70,71,72,73,74]. Epac (Exchange protein directly activated by cAMP) proteins are intracellular sensors for cAMP and mediate its effects to activate Rap1 [75,76]. The two isoforms include Epac1 which is ubiquitously expressed, with particularly high levels in the thyroid, among other tissues, and Epac2 which is not detected in the thyroid; however, it is expressed in a limited number of other tissues [76,77]. Epac regulates Rap activity in concert with and independently of PKA, and the effects—either stimulatory or inhibitory—seem to depend on the cellular context and the type of stimuli [69,75,78,79]. In addition, it has been shown that Epac1 plays a role in cell migration and invasion in other types of cancer [78,80]. Loss of *Rap*, specifically of the Rap1b isoform, in *Prkar1a* KO thyroids in the setting of overactivation of the PKA pathway, resulted in reduced risk of developing thyroid cancer by 65%; this occurred independently of Epac1 as its deletion did not have any effect in PKA-Rap1 associated thyroid tumorigenesis, underlying the essential role of PKA-Rap1 signaling in the development of FTC [68]. However, even though tumor suppression happened to a significant extent, the carcinogenic phenotype was not completely rescued, which led to the speculation that more complex signaling interactions may be involved [68].

These findings were further supported by other studies that showed that Rap proteins can be directly regulated by PKA using a specific phosphorylation site at serine 180 on Rap1a and serine 179 on Rap1b [81]. When PKA phosphorylates Rap, it regulates its subcellular localization, and its downstream effectors such as ERK and Rap-dependent regulation of cell migration [82,83]. These previous studies indicate that PKA can control Rap action and downstream cellular processes directly suggesting that PKA-Rap1 pathway is independent of Epac1 in thyroid cancer. On the other hand, previous studies have shown, that both PKA and Epac signal to Rap1 downstream of TSH [69,75], but it seems to be tissue-dependent [78,79].

In combination, these studies demonstrated that cAMP or PKA signaling or both play an important role in tumor development and that additional factors may contribute to *Prkar1a* haploinsufficiency in causing those tumors. *Trp53+/−* mice and other animal models for diseases like CNC, including Peutz–Jeghers and neurofibromatosis type 1, did not exhibit the same phenotype as in humans; it only occurred when one or more tumor suppressor genes were knocked out as well [84,85,86]. *Prkar1a* haploinsufficiency in addition to either *Trp53* or *Rb1* haploinsufficiency resulted in more tumors and decreased survival compared to *Trp53+/−* or *Rb1+/−* mice [87]. Specifically, *Rb1+/− Prkar1a+/−* mice developed more MTCs than *Rb1+/−* mice [87].

### 3.2. Studies in Humans

Further evidence to support the involvement of PKA in thyroid tumors was demonstrated by studying the *PRKAR1A* gene in thyroid tissue from patients with CNC [29]. The involvement of the thyroid in the syndrome was reported for the first time twelve years after CNC was first described [88], in 1997 [89]. In a cohort of 53 individuals with familial CNC, thyroid disease was identified in 11% of patients; of them, three were studied in detail, two with thyroid carcinomas (one PTC, one FTC) and one patient with a benign follicular adenoma. [89]. In addition, 60% of patients with the sporadic form of the complex exhibited thyroid gland lesions of follicular origin [89]. The authors concluded that thyroid carcinomas may develop in situ from precursor benign lesions in these patients. It is important to note that patients’ ethnicity does not seem to play a role in CNC phenotype that include thyroid carcinomas.

Since the *PRKAR1A* gene was identified as causal in CNC, many disease-causing mutations have been identified [24,90]. Sandrini et al. showed that in thyroid cancer the activity of PKA is greater than in adenomas, partly due to genetic defects in the *PRKAR1A* gene and/or locus [26]. The region 17q22–24 was frequently lost in cancer but not in benign tumors. In addition, it was shown that RIα, the most abundant regulatory subunit of cAMP-dependent PKA [91], in thyroid cells, possibly exhibits a tumor-suppressor function, as indicated by decreased expression of the RIα subunit in carcinomas and by the losses of *PRKAR1A* 17q22–24 locus in about 50% of all informative cancers. It has been known that the activation of cAMP/PKA pathway is involved in normal thyroid cell growth [92]; the same appears to be true for thyroid adenomas, while in the case of PTCs inhibition is induced [93]. The results suggested that *PRKAR1A* is indeed involved in sporadic thyroid tumors, along with other genes [94,95,96,97], some of which could be associated with the PKA pathway [91]. Any disruption of that, because of deficiency of the RIα subunit, could lead to cAMP-dependent PKA mediated cell proliferation and/or stimulation of other pathways linked to proliferation of thyroid cells [91,98].

In a recent series of 353 CNC patients from 185 families, patients from various ethnicities and with a wide range of clinical manifestations were studied [99]. More than 60% of them harbored mutations in the *PRKAR1A* gene. In 25% of patients, thyroid tumors were present while thyroid cancer (either FTC or PTC or both) was present in 2.5% of cases. In addition, thyroid tumors (*p* = 0.016) were more frequent in *PRKAR1A* carriers and presented at a younger age (*p* = 0.03). Moreover, they were more commonly associated with the ‘hot spot’ c.491–492delTG mutation in comparison with all other *PRKAR1A* defects. It was also observed that patients with no mutations of the *PRKAR1A* gene or its genomic locus 17q22–24, were less likely to develop thyroid tumors. In a review of 26 patients, in 61% of them benign lesions (including follicular adenoma, follicular hyperplasia or nodular hyperplasia) were detected, while 38% of them had thyroid carcinomas (seven with FTC and three with PTC). The majority of patients presented with an asymptomatic thyroid nodule and included middle-aged women [100].

## 4. Other Molecular Events in Thyroid Cancer

A significant number of mutations in thyroid cancer involves encoding genes of the MAPK and PI3K/AKT pathways. Mutated genes that affect these pathways encode the signal transduction molecules RAS, BRAF and NTRK1 and RET receptor tyrosine kinases. These mutations are present in approximately 70% of patients with PTCs and they exhibit particular clinical manifestations as well as specific histopathological characteristics in the tumor level [101,102,103,104]. Among FTCs, *RAS* mutations and *PAX8/PPARγ* rearrangements are common [105]. Because PAX8 is important for the development of the thyroid, it has been speculated that the fusion of *PPARγ* and *PAX8* can lead to cancer by activation of aberrant gene transcription [3]. *PAX8/PPARγ* is found in FTCs with a frequency of 30–35% and in a very small percentage of the follicular variant PTCs and follicular adenomas [106,107,108,109]. Thyroid tumors harboring *RAS* mutations, most commonly *NRAS* and *HRAS* mutations, include FTCs in 40–50%, PTCs in 10–20% and 20–40% of anaplastic and poorly differentiated carcinomas [110,111,112,113,114,115,116]. Because PTCs that harbor *RAS* mutations form neoplastic follicles and no papillary structures, they are known as follicular variant of PTC [101,117]. Benign follicular adenomas have also been found to harbor *RAS* mutations in 20–40%, indicating that they may be precursors of *RAS*-positive FTCs and follicular variant of PTCs [118,119,120,121]. Furthermore, BRAF^V600E^ mutation represents 98–99% of all BRAF mutations in thyroid cancer [122,123,124]. It accounts for 40–45% of classic PTCs, 30–40% of ATCs, and 20–40% of poorly differentiated thyroid carcinomas [125,126,127,128].

Mutations in *BRAF* and *RAS* are thought to represent an early event in the progression of thyroid cancer, given that they are present in poorly and well-differentiated thyroid cancer, as well as in ATCs. On the other hand, additional genetic alterations are usually present in poorly differentiated carcinomas and ATCs; these constitute late events that may be necessary for tumor dedifferentiation. These genetic alterations include mutations in the *TP53* and *CTNNB1* genes and encoding genes of the PI3K/AKT signaling pathway [105]. In 50–80% of ATCs, point mutations that lead to loss-of-function of *TP53* have been identified; these are very rare in well-differentiated thyroid cancer [129,130]. *CTNNB1* mutations occur in approximately 60% of ATCs [131,132].

## 5. Medullary Thyroid Cancer as Part of MEN2 Syndromes

About one-third of MTCs are hereditary, presenting as multicentric and bilateral, in contrast with sporadic cases that are a single unilateral tumor [133,134]. They present as part of MEN2A (70–80%), MEN2B (5%), or familial MTC (FMTC) (10–20%). The first inherited subtype of MTC, MEN2A, consists of primary hyperparathyroidism, pheochromocytoma and MTC in which it can occur early in life (approximately 5 years of age) in contrast with sporadic cases that presents between 15 and 20 years [134,135]. MEN2B is characterized by pheochromocytoma, MTC and non-endocrine diseases such as mucosal neuromas, intestinal tumors (most commonly ganglioneuromas) and Marfanoid habitus [135]. In the case of FMTC, only the thyroid gland is affected, but in a significant number of relatives in the same family, usually between the ages of 20 and 40 [135,136,137]. Activating germline *RET* mutations have been identified as the main cause of up to 98% of hereditary MTCs and up to half of sporadic cases [138]. Depending on the mutated residue within the RET protein, the phenotype may differ [139,140,141,142]. Families with two or more members with MTC are referred for genetic counseling and screening, if positive they undergo further testing for hyperparathyroidism and pheochromocytoma [2,143,144]. In the case of sporadic MTCs, somatic *RET* mutations, particularly M918T, has been shown to be associated with more aggressive disease and worse prognosis [144,145].

## 6. Anaplastic Thyroid Carcinoma

ATC is a rare (1–2%) but very aggressive type of thyroid cancer [146] with average age at diagnosis over 70 years [147]. It is considered to evolve from dedifferentiation of a pre-existing DTC caused by accumulation of several genetic alterations that lead to disruption of two signaling pathways that are involved in cell proliferation, PI3K-AKT and MAPK [148,149,150]. The most common mutations include *TP53*, which is considered a genetic hallmark of ATC, as well as *RAS*, *BRAF*, *PIK3CA* [151,152], mutations that have also been identified in DTC [153]. Median survival is usually less than 6 months after diagnosis and the mortality rate is >90% [154,155]. Due to its extremely aggressive nature, it is critical to be diagnosed promptly. Clinical symptoms are usually used for the diagnosis, in contrast with DTC in which diagnosis is made by FNA of a suspicious nodule [147]. The symptoms can last from 4 weeks to 11 months and usually consist of a rapidly enlarging neck mass along with vocal cord paralysis and dyspnea [147].

## 7. Systemic Treatments for Thyroid Cancer

Two multikinase inhibitors (MKI), lenvatinib and sorafenib, are currently approved by the US Food and Drug Administration (FDA) for the treatment of advanced DTC. Sorafenib was approved based on the favorable results of a placebo-controlled phase 3 clinical trial (DECISION) [156]. The positive results of the lenvatinib phase 3 SELECT trial [157] as well as a phase 2 study led to the approval of that drug [158]. Cabozatinib and vandetanib are approved by the FDA for the treatment of MTC. Vandetanib is approved for symptomatic, unresectable, locally advanced, or metastatic MTC in patients based on a phase 3 trial (ZETA) [159]. Cabozantinib was studied in a phase 3 clinical trial (EXAM) [160] and showed good results while another clinical trial in MTC patients is still active (EXAMINER, NCT01896479). RET-inhibitors have been studied as well for thyroid cancers that harbor *RET* mutations (NCT03157128, NCT04211337, NCT03906331, NCT04280081, NCT03037385).

### 7.1. Immunotherapy

In the recent years, immunotherapy has emerged as a new transformative approach into the body’s natural antitumor defenses. To date, there is no approved immunotherapy for advanced thyroid cancer. A few clinical trials using novel immunotherapy agents like programmed cell death protein 1 (PD-1) checkpoint inhibitors are ongoing. Pembrolizumab in an Ib phase trial (KEYNOTE) showed a tumor size reduction of 35–50% in PTC and FTC. The use of another anti-PD1 agent (spartalizumab) was evaluated in progressive ATC that responded to therapy [161]. In an ongoing phase 2 clinical trial (NCT03246958), the efficacy of the combination of nivolumab (anti-PD1-1) and ipilimumab (anti-CTLA-4- cytotoxic T-lymphocyte-associated protein 4) was evaluated in patients with aggressive thyroid cancer. In addition, multiple clinical trials with VEGF and/or VEGF inhibitor and immune checkpoint inhibitors have been designed. Pemproblizumab plus lenvatinib was investigated in a phase 2 trial for unresectable ATC (NCT04171622) as well as in a randomized study in a small group of advanced ATC and PDTC [162]. The same combination is under study in DTC and PDTC naïve or progressing after lenvatinib patients (NCT02973997). Triple combined therapy (cabozantinib plus nivolumab and ipilimumab) is under evaluation for DTC and PDTC (NCT03914300).

### 7.2. Treatment for PRKAR1A-Associated Thyroid Tumors

To date, there is no medical treatment targeting cAMP/PKA signaling in CNC. Surgical treatment is the treatment of choice in patients with PRKAR1A-associated thyroid tumor [163].

## 8. Clinical Surveillance in Patients with PRKAR1A-Associated Thyroid Tumors

Human studies in CNC underly the importance of investigating thyroid nodules in these patients. Multiple thyroid nodules are present in up to 75% of patients with CNC on thyroid ultrasound; the majority of them are non-functioning follicular adenomas [164]. However, thyroid carcinomas are common as well. Early detection is vital and CNC patients should be followed with long-term clinical and/or ultrasound surveillance with biopsy of suspicious nodules, for early detection of carcinomas [164].

Because CNC is inherited in an autosomal dominant manner, each child of an affected individual has a 50% chance of inheriting the pathogenic variant. Most of the affected patients (approximately 70%) have an affected parent. In the case that the pathogenic variant is known in a family, prenatal testing may be recommended [164].

## 9. Conclusions

In summary, recent advances in molecular mechanisms of thyroid cancer have improved cancer prognosis and detection. *PRKAR1A*, a regulator of PKA activity, is possibly involved in the molecular events that contribute to thyroid cancer. Identifying the genetic basis of *PRKAR1A*-associated thyroid tumors is important as it will provide better clinical management to these patients.

## Figures and Tables

**Figure 1 cancers-13-03834-f001:**
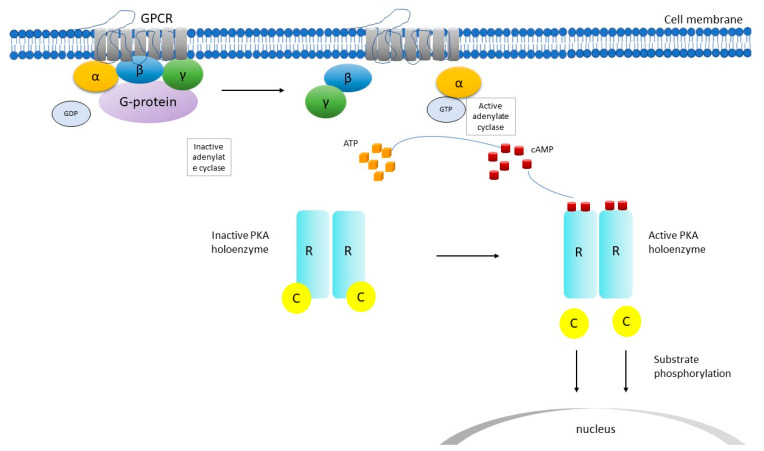
Schematic representation of cyclic adenosine monophosphate (cAMP) signaling pathway. *C* catalytic subunit of PKA, *GDP* guanosine diphosphate, *GPCR* G-protein coupled receptor, *GTP* guanosine triphosphate, *PKA* protein kinase, *R* regulatory subunit of PKA, *α*, *β*, *γ* subunits. After the GPCR is activated, adenylate cyclase is activated and produces cAMP, which binds to the R subunit and activates PKA. Then, conformational changes ensue and the C subunits are released and phosphorylate cytoplasmic targets.

**Table 1 cancers-13-03834-t001:** Diagnostic criteria for Carney Complex [24].

Main Diagnostic Criteria
Spotty skin pigmentation with a typical distribution (lips, conjunctiva and inner or outer canthi, vaginal and penile mucosa)
2. Myxoma (cutaneous and mucosal) ^a^
3. Cardiac myxoma ^a^
4. Breast myxomatosis ^a^ or fat-suppressed magnetic resonance imaging findings suggestive of this diagnosis
5. PPNAD ^a^ or paradoxical positive response of urinary glucocorticosteroids to dexamethasone administration during 6-day modified Liddle test
6. Acromegaly due to GH-producing adenoma ^a^
7. LCCSCT ^a^ or characteristic calcification on testicular ultrasonography
8. Thyroid carcinoma ^a^ or multiple, hypoechoic nodules on thyroid ultrasonography
9. Psammomatous melanotic schwannoma ^a^
10. Blue nevus, epithelioid blue nevus (multiple) ^a^
11. Breast ductal adenoma (multiple)
12. Osteochondroma of bone ^a^
**Supplementary Criteria**
Affected 1st-degree relative
2. Inactivating mutation of the *PRKAR1A* gene

^a^ with histologic confirmation; *LCCSCT* large cell calcifying Sertoli cell tumor, *PPNAD* primary pigmented nodular adrenocortical disease; To make a diagnosis of CNC, a patient must either: (1) exhibit two of the manifestations of the disease listed, or (2) exhibit one of these manifestations and meet one of the supplemental criteria.

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
