# Peer review of "PRKAR1A and Thyroid Tumors"

_cancers, 2021, doi:10.3390/cancers13153834_

Round 1

Reviewer 1 Report

The authors present a comprehensive and yet concise review of PRKAR1A-associated signaling in thyroid cancer in both experimental and clinical aspects. 

minor comment

If the author could add a section on non-invasive detection of thyroid tumors, it will make this review even more complete. In addition, potential drugs targeting PRKAR1A-associated signaling molecules would be of importance to the readers.

Author Response

Response to Reviewer 1 [R1] comments:

[R1]: The authors present a comprehensive and yet concise review of PRKAR1A-associated signaling in thyroid cancer in both experimental and clinical aspects.

[A]: We thank the reviewer for the positive comments.

  1. [R1]: minor comment. If the author could add a section on non-invasive detection of thyroid tumors, it will make this review even more complete. In addition, potential drugs targeting PRKAR1A-associated signaling molecules would be of importance to the readers.

[A]: Dear R1 thank you for pointing this out. We added two sections to address these points:

“Thyroid nodules are quite common and are found either clinically or as an incidental finding on imaging studies [8]. The majority of them are benign [9]; only a small percent harbors thyroid cancer [8]. The initial steps in the evaluation of a thyroid nodule consist of medical history including symptoms (recent onset of hoarseness, neck discomfort or dysphagia), history of head/neck radiation and personal/family history of cancer, followed by physical examination and measurement of serum thyrotropin levels. Ultrasonography (US) is the next step in order to determine the size of the nodule, its characteristics and to assess for cervical lymphadenopathy [10]. If thyrotropin levels are normal or elevated and the nodule size is >1 cm, then fine needle aspiration (FNA) is indicated, according to the American Thyroid Association guidelines [11,12]. If thyrotropin levels are low, then Iodine-123 or technetium-99m thyroid scanning is recommended. In the case that the nodule is nonfunctioning and bigger than 1cm, FNA is the next step. If the cytological interpretation is benign, then repeated FNA is not required unless suspicious features appear in the follow up [12,13]. Currently, US-guided FNA is the gold standard in the diagnosis; however, in about 25% of the cases, the diagnosis remains indeterminate [9,14-20]. If cytologic results are interpreted as atypia of underdetermined significance or follicular lesion of underdetermined significance, then in the case of high suspicion, assessment of the aspirate for molecular abnormalities (eg mutations or rearrangements) is indicated [21].” (page 2, lines 73-92 in the tracking version or lines 68-88 in the yellow highlight version).

In addition, we added a new section about thyroid treatment: “Two multikinase inhibitors (MKI), lenvatinib and sorafenib, are currently approved by the US Food and Drug Administration (FDA) for the treatment of advanced DTC. Sorafenib was approved based on the favorable results of a placebo-controlled phase 3 clinical trial (DECISION) [156]. The positive results of the lenvatinib phase 3 SELECT trial [157] as well as a phase 2 study led to the approval of that drug [158]. Cabozatinib and vandetanib are approved by the FDA for the treatment of MTC. Vandetanib is approved for symptomatic, unresectable, locally advanced, or metastatic MTC in patients based on a phase 3 trial (ZETA) [159]. Cabozantinib was studied in a phase 3 clinical trial (EXAM) [160] and showed good results while another clinical trial in MTC patients is still active (EXAMINER, NCT01896479). RET-inhibitors have been studied as well for thyroid cancers that harbor RET mutations (NCT03157128, NCT04211337, NCT03906331, NCT04280081, NCT03037385).

7.1. Immunotherapy

In the recent years, immunotherapy has emerged as a new transformative approach into the body’s natural antitumor defenses. To date, there is no approved immunotherapy for advanced thyroid cancer. A few clinical trials using novel immunotherapy agents like programmed cell death protein 1 (PD-1) checkpoint inhibitors are ongoing. Pembrolizumab in an Ib phase trial (KEYNOTE) showed a tumor size reduction of 35-50% in PTC and FTC. The use of another anti-PD1 agent (spartalizumab) was evaluated in progressive ATC that responded to therapy [161]. In an ongoing phase 2 clinical trial (NCT03246958), the efficacy of the combination of nivolumab (anti-PD1-1) and ipilimumab (anti-CTLA-4- cytotoxic T-lymphocyte-associated protein 4) was evaluated in patients with aggressive thyroid cancer. In addition, multiple clinical trials with VEGF and/or VEGF inhibitor and immune checkpoint inhibitors have been designed. Pemproblizumab plus lenvatinib was investigated in a phase 2 trial for unresectable ATC (NCT04171622) as well as in a randomized study in a small group of advanced ATC and PDTC [162]. The same combination is under study in DTC and PDTC naïve or progressing after lenvatinib patients (NCT02973997). Triple combined therapy (cabozantinib plus nivolumab and ipilimumab) is under evaluation for DTC and PDTC (NCT03914300).

7.2. Treatment for PRKAR1A-associated thyroid tumors

To date, there is no medical treatment targeting cAMP/PKA signaling in CNC. Surgical treatment is the treatment of choice in patients with PRKAR1A-associated thyroid tumor [163].”  (pages 9-10, lines 517-571 in the tracking version or lines 357-403 in the yellow highlight version).

Reviewer 2 Report

This is a timely review of thyroid cancer with a particular focus on the cAMP-dependent protein kinase A (PKA) type 1α regulatory subunit encoded by the PRKARIA gene whose inactivation is associated with an unchecked activity of PKA and thyroid cancer. The authors describe several mouse models and human studies with Carney Complex (CNC) syndrome patients who carry an inactivating mutation in the PRKARIA gene and exhibit increased incidence of thyroid tumorigenesis. CNC is associated with a number of tumor types, including follicular (FTC) and papillary (PTC), mostly well differentiated and most common of all thyroid cancers subtypes.  Despite the importance of the link between CNC and thyroid cancer, there are concerns about the organization of the manuscript, its dated references, and lack of any information regarding clinical trials for thyroid cancer. While the focus of the review is on PRKARIA in FTC, the authors include a short section on medullary thyroid cancer, but ignore the anaplastic thyroid carcinoma, an aggressive subtype associated with poor prognosis.  Furthermore, it would be important to present a concise summary of more recent molecular insights into thyroid cancer biology, as well as novel therapeutic strategies, including immunotherapy.  Lastly, the manuscript should be edited for grammar.

Author Response

Response to Reviewer 2 [R2] comments:

  1. [R2]: This is a timely review of thyroid cancer with a particular focus on the cAMP-dependent protein kinase A (PKA) type 1α regulatory subunit encoded by the PRKARIA gene whose inactivation is associated with an unchecked activity of PKA and thyroid cancer. The authors describe several mouse models and human studies with Carney Complex (CNC) syndrome patients who carry an inactivating mutation in the PRKARIA gene and exhibit increased incidence of thyroid tumorigenesis. CNC is associated with a number of tumor types, including follicular (FTC) and papillary (PTC), mostly well differentiated and most common of all thyroid cancers subtypes.Despite the importance of the link between CNC and thyroid cancer, there are concerns about the organization of the manuscript, its dated references, and lack of any information regarding clinical trials for thyroid cancer. While the focus of the review is on PRKARIA in FTC, the authors include a short section on medullary thyroid cancer, but ignore the anaplastic thyroid carcinoma, an aggressive subtype associated with poor prognosis.

[A]: Dear Reviewer 2, thank you for pointing this out. The references for PRKAR1A and thyroid are dated because there has not been anything newer in the field. Part of our impetus to publish this review is to reinvigorate research on PRKAR1A and thyroid cancer.

We added a paragraph about anaplastic thyroid carcinoma: “ATC is a rare (1-2%) but very aggressive type of thyroid cancer [146] with average age at diagnosis over 70 years [147]. It is considered to evolve from dedifferentiation of a pre-existing DTC caused by accumulation of several genetic alterations that lead to disruption of two signaling pathways that are involved in cell proliferation, PI3K-AKT and MAPK [148-150]. The most common mutations include TP53, which is considered a genetic hallmark of ATC, as well as RAS, BRAF, PIK3CA [151,152]; mutations that have also been identified in DTC [153]. Median survival is usually less than 6 months after diagnosis and the mortality rate is >90% [154,155]. Due to its extremely aggressive nature, it is critical to be diagnosed promptly. Clinical symptoms are usually used for the diagnosis, in contrast with DTC in which diagnosis is made by FNA of a suspicious nodule [147]. The symptoms can last from 4 weeks to 11 months and usually consist of a rapidly enlarging neck mass along with vocal cord paralysis and dyspnea [147].” (page 9, lines 518-529 in the tracking version or lines 358-369 in the yellow highlight version).

2. [R2]: Furthermore, it would be important to present a concise summary of more recent molecular insights into thyroid cancer biology, as well as novel therapeutic strategies, including immunotherapy.  Lastly, the manuscript should be edited for grammar.

[A]: Thank you for your comment. We looked more carefully trying to fix grammar errors and a new section was added to address your comment:

“Two multikinase inhibitors (MKI), lenvatinib and sorafenib, are currently approved by the US Food and Drug Administration (FDA) for the treatment of advanced DTC. Sorafenib was approved based on the favorable results of a placebo-controlled phase 3 clinical trial (DECISION) [156]. The positive results of the lenvatinib phase 3 SELECT trial [157] as well as a phase 2 study led to the approval of that drug [158]. Cabozatinib and vandetanib are approved by the FDA for the treatment of MTC. Vandetanib is approved for symptomatic, unresectable, locally advanced, or metastatic MTC in patients based on a phase 3 trial (ZETA) [159]. Cabozantinib was studied in a phase 3 clinical trial (EXAM) [160] and showed good results while another clinical trial in MTC patients is still active (EXAMINER, NCT01896479). RET-inhibitors have been studied as well for thyroid cancers that harbor RET mutations (NCT03157128, NCT04211337, NCT03906331, NCT04280081, NCT03037385).

7.1. Immunotherapy

In the recent years, immunotherapy has emerged as a new transformative approach into the body’s natural antitumor defenses. To date, there is no approved immunotherapy for advanced thyroid cancer. A few clinical trials using novel immunotherapy agents like programmed cell death protein 1 (PD-1) checkpoint inhibitors are ongoing. Pembrolizumab in an Ib phase trial (KEYNOTE) showed a tumor size reduction of 35-50% in PTC and FTC. The use of another anti-PD1 agent (spartalizumab) was evaluated in progressive ATC that responded to therapy [161]. In an ongoing phase 2 clinical trial (NCT03246958), the efficacy of the combination of nivolumab (anti-PD1-1) and ipilimumab (anti-CTLA-4- cytotoxic T-lymphocyte-associated protein 4) was evaluated in patients with aggressive thyroid cancer. In addition, multiple clinical trials with VEGF and/or VEGF inhibitor and immune checkpoint inhibitors have been designed. Pemproblizumab plus lenvatinib was investigated in a phase 2 trial for unresectable ATC (NCT04171622) as well as in a randomized study in a small group of advanced ATC and PDTC [162]. The same combination is under study in DTC and PDTC naïve or progressing after lenvatinib patients (NCT02973997). Triple combined therapy (cabozantinib plus nivolumab and ipilimumab) is under evaluation for DTC and PDTC (NCT03914300).

7.2. Treatment for PRKAR1A-associated thyroid tumors

To date, there is no medical treatment targeting cAMP/PKA signaling in CNC. Surgical treatment is the treatment of choice in patients with PRKAR1A-associated thyroid tumor [163].”  (pages 9-10, lines 517-571 in the tracking version or lines 357-403 in the yellow highlight version).

Reviewer 3 Report

Authors summarized published data regarding the role of RRKAR1A in thyroid tumors. A comprehensive and extensive literature review was carried out. 

Author Response

Response to Reviewer 3 [R3] comments:

1.[R3]: Authors summarized published data regarding the role of RRKAR1A in thyroid tumors. A comprehensive and extensive literature review was carried out.

[A]: We thank the reviewer for the positive comments.

Reviewer 4 Report

Title: PRKAR1A and thyroid tumors

Journal: Cancers (ISSN 2072-6694)

Correspondence: Fabio R. Faucz; fabio.faucz@nih.gov

Major point:
In general the review is interesting and well writing, but the 
The chapter two that correspond to: PRKAR1A structure and function (line 90) should be redo,  because it is not enough clear and including the squeme of the gene. Inb this sense, the authors should be  explain better the pathway of the Figure 1 adn redo this figure.
Moreover,  the authors shoud be including  which stimuli shoud be necessaries and their importance of its (line 103) 
MInor points:
LIne 139.- they authors should be Indicated the type of thyroid neoplasm are present in the mutated mice 
LIne 154.- Check the number of the bibliography
LIne 253.- Indicated the type of ethnicities and also if it is relevant this data

Author Response

Response to Reviewer 4 [R4] comments:

1.[R4]: Major point: In general the review is interesting and well writing, but the chapter two that correspond to: PRKAR1A structure and function (line 90) should be redo, because it is not enough clear and including the squeme of the gene. In this sense, the authors should be explain better the pathway of the Figure 1 and redo this figure. Moreover,  the authors should be including  which stimuli should be necessaries and their importance of its (line 103)

[A]: Dear Reviewer 3, thank you for your comment. Based on your recommendations, section 2 was redone and Figure 1 was explained in more detail:

“Cyclic adenosine monophosphate (cAMP)-dependent protein kinase type 1-alpha regulatory subunit is encoded by the PRKAR1A gene. PRKAR1A consists of 11 exons; ten of them (2-11) are coding. Protein kinase A (PKA) (Figure 1), a serine/threonine kinase, is a second messenger-dependent enzyme and it is involved in G-protein coupled intracellular pathways. It is the main mediator of cAMP actions for various cellular processes in mammals, including cell differentiation, proliferation and apoptosis [30-32].

The PKA holoenzyme is a hetero-tetramer composed of two regulatory (R) subunits and each is bound to one catalytic (C) subunit [33]. Four subtypes of R (RIα, RIβ, RIIα, RIIβ) and four subtypes of C (Cα, Cβ, Cγ and Prkx) subunits have been identified so far. A gene is coding each R (PRKR1A, PRKR1B, PRKR2A, PRKR2B) and each C (PRKACA, PRKACB, PRKACG, PRKX) subunit, respectively [33,34]. Two major isozymes have been identified, type I and type II PKA, based on their chromatographic elution patterns [32]; they are comprised of homodimers of either RIα and RIβ or RIIα and RIIβ respectively [31,35]. In the basal state, the catalytic subunits bind mostly to type II subunits [31,35-37]. When cAMP binds to the R subunits, it alters their conformation; this causes the dissociation of each active C subunit from the dimer with the corresponding R subunit. Following that, the free C subunits phosphorylate threonine and serine residues of proteins that are critical to the activation of downstream processes [38-40].” (page 4, lines 142-160 in the tracking version or lines 116-137 in the yellow highlight version).

“Figure 1. Schematic representation of cyclic adenosine monophosphate (cAMP) signaling pathway. C catalytic subunit of PKA, GDP guanosine diphosphate, GPCR G-protein coupled receptor, GTP guanosine triphosphate, PKAprotein kinase, R regulatory subunit of PKA, α,β,γ subunits. After the GPCR is activated, adenylate cyclase is activated and produces cAMP, which binds to the R subunit and activates PKA. Then, conformational changes ensue and the C subunits are released and phosphorylate cytoplasmic targets”. (page 4, lines 161-167 in the tracking version or lines 138-144 in the yellow highlight version).

2.[R4]: Minor points:
Line 139.- the authors should be indicated the type of thyroid neoplasm are present in the mutated mice

[A]: We mentioned that (thyroid follicular hyperplasia and adenomas) in line 217 in the new “tracking” version (or line 160 in the yellow highlight version)

Line 154. - Check the number of the bibliography

[A]: Thank you for pointing this out. The number of the bibliography is correct.

Line 253.- Indicated the type of ethnicities and also if it is relevant this data

[A]: We thank the reviewer for the comment. All patients in the referred manuscript were mixed American Caucasian. However, we do not have access at the clinical records anymore and it would be impossible to check if it is true what the patient said regarding their ethnicity during the anamnesis. So, we chose not to include this information in the manuscript. Following the second part of the question, we can also confirm that the ethnicity background is not relevant in the Carney complex patients’ phenotype, including thyroid problems. We included the following sentence in the manuscript: “Important to note that patients’ ethnicity does not seem to play a role in CNC phenotype that include thyroid carcinomas”. (page 7, lines 378-379 in the tracking version or lines 277-278 in the yellow highlight version).
